# Mosquito-Associated Viruses and Their Related Mosquitoes in West Africa

**DOI:** 10.3390/v13050891

**Published:** 2021-05-12

**Authors:** Eric Agboli, Julien B. Z. Zahouli, Athanase Badolo, Hanna Jöst

**Affiliations:** 1Molecular Biology and Immunology Department, Bernhard Nocht Institute for Tropical Medicine, 20359 Hamburg, Germany; agboli@bnitm.de; 2Department of Epidemiology and Biostatistics, School of Public Health, University of Health and Allied Sciences, Ho PMB 31, Ghana; 3Centre d’Entomologie Médicale et Vétérinaire, Université Alassane Ouattara, Bouake, 27 BP 529 Abidjan 27, Cote D’Ivoire; julien.zahouli@csrs.ci; 4Centre Suisse de Recherches Scientifiques en Côte d’Ivoire, Département de Recherche et Développement, 01 BP 1303 Abidjan 01, Cote D’Ivoire; 5Department of Epidemiology and Public Health, Swiss Tropical and Public Health Institute, 4051 Basel, Switzerland; 6Laboratory of Fundamental and Applied Entomology, Universitée Joseph Ki-Zerbo, Ouagadougou 03 BP 7021, Burkina Faso; a.badolo@gmail.com; 7Bernhard Nocht Institute for Tropical Medicine, WHO Collaborating Centre for Arbovirus and Hemorrhagic Fever Reference and Research, 20359 Hamburg, Germany

**Keywords:** mosquito-associated viruses, mosquito-borne viruses, mosquito-specific viruses, mosquitoes, urbanisation, land use change, climate change, West Africa

## Abstract

Mosquito-associated viruses (MAVs), including mosquito-specific viruses (MSVs) and mosquito-borne (arbo)viruses (MBVs), are an increasing public, veterinary, and global health concern, and West Africa is projected to be the next front for arboviral diseases. As in-depth knowledge of the ecologies of both western African MAVs and related mosquitoes is still limited, we review available and comprehensive data on their diversity, abundance, and distribution. Data on MAVs’ occurrence and related mosquitoes were extracted from peer-reviewed publications. Data on MSVs, and mosquito and vertebrate host ranges are sparse. However, more data are available on MBVs (i.e., dengue, yellow fever, chikungunya, Zika, and Rift Valley fever viruses), detected in wild and domestic animals, and humans, with infections more concentrated in urban areas and areas affected by strong anthropogenic changes. *Aedes aegypti*, *Culex quinquefasciatus*, and *Aedes albopictus* are incriminated as key arbovirus vectors. These findings outline MAV, related mosquitoes, key knowledge gaps, and future research areas. Additionally, these data highlight the need to increase our understanding of MAVs and their impact on host mosquito ecology, to improve our knowledge of arbovirus transmission, and to develop specific strategies and capacities for arboviral disease surveillance, diagnostic, prevention, control, and outbreak responses in West Africa.

## 1. Introduction

Mosquito-associated viruses (MAVs) include mosquito-specific viruses (MSVs) and mosquito-borne (arbo)viruses (MBVs). MBVs and their vectors (e.g., *Aedes*, *Culex* etc.) are an increasing public, veterinary, and global health concern [1,2,3,4]. Africa is known as the reservoir and origin of most pathogenic and nonpathogenic MAVs spreading around the globe. MAVs have a vast impact on livestock [5,6,7] and public and global health [1,2,3]. MBVs, including dengue virus (DENV), yellow fever virus (YFV), chikungunya virus (CHIKV), Zika virus (ZIKV), Rift Valley fever virus (RVFV), and West Nile virus (WNV), are increasingly threatening over 831 million (70%) people on the African continent [2,8], with a high risk of international spread due to globalisation [1,3]. Indeed, with high species diversity estimated at 3500 species, a number of mosquitoes (family *Culicidae*) are known to be efficient hosts, carriers, or vectors of various pathogens, including viruses that can infect animals and humans [1,2,3].

Mosquito-associated microbiota, comprising MSVs, bacteria, fungi, and protozoa, can profoundly influence many host phenotypes, including vector competence, which can either be enhanced or suppressed [9,10,11,12,13,14,15]. Therefore, MSVs represent promising tools and platforms for the development of vaccines against arboviruses infecting domestic animals (e.g., horses, pigs, etc.) and humans [16,17]. Production of such MSV-based vaccines can be performed at a lower biosafety level, making the process cheaper and safer [16,18,19]. Additionally, some of the MSVs (e.g., flaviviruses, mononegaviruses, totiviruses, etc.) can affect the physiology, biology, ecology, evolution, diversity, and abundance of mosquito hosts, and thus act as biological agents of vector control by reducing, preventing, or inhibiting the transmission of pathogens (e.g., malaria *Plasmodium* and arboviruses) [4,12,13,14]. However, certain microbiomes, including MSVs (e.g., flaviviruses, alphaviruses, bunyaviruses, etc.), colonise the mosquito midgut, and can alter, degrade, or detoxify ingested insecticides, and thus increase mosquito vector resistance to insecticides and its lifespan, which can lead to a high risk of transmission of pathogens [20,21]. 

MBVs, such as RVFV, can infect and cause abortion in humans and domestic animals (e.g., ruminants), and cause important economy losses [6,7,22,23], and others, including YFV, DENV, CHIKV, ZIKV, and WNV, are responsible for millions of disease cases and several thousands of deaths among humans [2]. However, there remain large gaps in our understanding of the current distribution of these arboviral diseases as vector-borne research and control in the region is focused primarily on *Anopheles* mosquitoes, *Plasmodium*, and malaria disease [2,3]. With an increase in incidence coupled with the lack of effective drugs, prophylaxis, and licensed commercial vaccines (except for yellow fever) for most of these arboviral infections and diseases, significant outbreaks of these diseases levy a substantial burden on global health and economics in various African countries [2,3]. West Africa is already highly vulnerable to arboviral outbreaks (27,000 arboviral cases since 2017) [3] and is expected to be the next front for arboviral diseases (i.e., dengue) [24].

MAVs in West Africa have been reviewed in the past by a few authors who focused on the field of MBVs only [1,2,3]. However, MAVs (i.e., MSVs and MBVs) are often discovered and their continual surveillance is key to harnessing effective control and emergency preparedness. West Africa has high floristic and faunistic biodiversity characterised by a diversified vegetation (e.g., forest and savannah) and land-cover change (e.g., agricultural, rural, and urban areas), wild and domestic vertebrate animals (e.g., reptiles, birds, mammals, etc.), mosquitoes (e.g., *Aedes*, *Anopheles*, *Culex*, etc.), and pathogens (i.e., MSVs and MBVs) [1,2,3,25]. Biological interactions among and between MAVs, mosquitoes, and vertebrate hosts coexist and evolve across different bioclimatic ecozones (i.e., arid in Sahel savannah over drier in Sudan savannah to moderately wetter Guinean and wetter Guineo–Congolian forest) [3,25,26,27]. Moreover, the western African region is particularly experiencing devastating effects of deforestation, agricultural encroachment, rapid and unplanned urbanisation, and climate change that can influence the ecologies of MAVs, hosts, and *Aedes* mosquitoes [3,25,26,27,28,29,30,31,32,33]. Therefore, it is imperative to review the common MAVs in the West African region, and provide a guide for emerging and re-emerging viruses. In this review, we document the diversity, abundance, and distribution of MAVs (i.e., MSVs and MBVs) of veterinary and medical interests, and the mosquito hosts, mosquito vectors, and vertebrate hosts involved in their life cycles and transmission cycles (i.e., sylvatic or zoonotic, emergence and urban or epidemic cycles) in the region. We examined also how urbanisation, land use, and global weather change can influence the epidemiology of MAVs and MSVs and the ecology of their vectors in West Africa. The findings increase our understanding of MAVs of medically important vectors and identify key knowledge gaps. We also identify future research areas to explore in order to improve and design strategies and policy for the biosurveillance and early detection of arboviruses for the prediction and prevention of arboviral outbreaks and the biocontrol of their mosquito vectors in West Africa.

## 2. Materials and Methods

### 2.1. General Search Strategy

In this review we systematically compiled the available published literature on MAVs in West Africa. The search was conducted in PubMed and Google Scholar by searching the terms ‘arbovirus’, ‘mosquito-borne arboviruses’, ‘mosquito-specific viruses’; ‘arboviral disease’ AND <country name> OR ‘West Africa’. Sources included also previous reviews about arboviruses and mosquito-specific viruses in Africa and other parts of the world [1,2,3,4]. Searches were not limited by year, going back to publication in the 1950s and until April 2021. Bibliographies of selected publications were also searched for supplementary reports. English, French, and Russian reports were screened. Publications were searched for data about virus isolation from humans, vertebrates, and mosquitoes, detected by reverse transcription polymerase chain reaction (RT-PCR), virus isolation in cell culture, inoculation in laboratory animals, and plaque reduction neutralization test (PRNT), and also for detections of IgM and IgG in humans and vertebrates by enzyme-linked immunosorbent assay (ELISA), rapid diagnostic tests (RDTs) based on immunochromatographic detection, and complement fixation test (CT). If available, ´year of virus detection´ and not ´year of publication´ were assigned in Table 1.

### 2.2. Trend Analysis of Mosquito-Associated Viruses

The figure was generated by using the syntax ‘mosquito-borne viruses, West Africa’, and ‘mosquito specific viruses, West Africa’ at NCBI database (date of search: 8 December 2020). The automatically generated CSV files showing results by year were pulled and plotted. The data in Figure 1 is not showing the number of discoveries but a combination of viruses discovered and number of times the statements ‘mosquito-borne viruses’ and ‘mosquito specific viruses’ were mentioned in literature.

## 3. Discovery of Mosquito-Associated Viruses

A cumulative number of MAVs are being discovered in hematophagous arthropods all over the world. There is a potential impact on the fitness and competence of the vector. Therefore, more research is needed to expand the knowledge about their prevalence, distribution, and diversity. Figure 1 describes the trend of MAVs reported in the literature. This trend enlightens how researches about MAVs are on the increase. The past decades have shown a dramatic surge in the literature recounting novel MSVs and classical arboviruses. The increase in research interest has provided new understanding about viral diversity and evolution. Finally, the new viruses discovered have created curiosity in their use as potential biological control agents. The distribution and detection of MAVs are visualised on the maps of Figure 2. 

### 3.1. Flaviridae

Flaviviruses are enveloped, spherical, and about 50 nm in diameter. The surface proteins of the viral capsid are arranged in an icosahedral-like symmetry. They have a monopartite, linear, ssRNA(+) genome of about 9.7–12 kb [147]. These viruses can be divided into three main groups: those that are maintained in an arthropod–vertebrate cycle and strictly depend on each other are called dual-host flaviviruses [148]; those that are limited to vertebrates with an unknown arthropod relationship are known as vertebrate-specific flavivirus [149]; and those that replicate only in insect cell lines are termed insect-specific flaviviruses [148,150,151]. Flaviviruses and flavivirus-specific antibodies detected in West Africa are shown in Figure 2.

The application of next generation sequencing (NGS) techniques in virus discovery has led to the identification and isolation of numerous MSVs [4]. West Africa is not left out in this discovery, and MAVs known to cause outbreaks and those which are not associated with human infections are outlined in Table 1. The first mosquito-specific flavivirus isolated was cell fusing agent virus (CFAV) in 1975 [152], but this virus was never identified in West Africa until 2016 in Ghana [131]. CFAV was isolated from mosquitoes collected via a mosquito surveillance field work from 2015 to 2016 in Ghana. Examples of other MSVs identified in West Africa include: Nounane virus and Nienokoue virus (Côte d’Ivoire); Anopheles flavivirus-like virus and Barkedji virus (Senegal); and Culex flavivirus (Ghana) [13,80,131,132,133]. Nounane and Nienokoue viruses were identified in Tai National Park, a tropical rainforest zone, from the *Culex* and *Uranotaenia* mosquitoes respectively [132,133].

The presence of flaviviruses in West Africa is not limited to MSVs, but also medically important MBVs, such as DENV, WNV, ZIKV, YFV, and Usutu virus (USUV). McNamara et al. seem to provide the first report on mosquito-borne flaviviruses, including YFV, WNV, ZIKV, and Uganda S virus in the 1950s [83]. West Africa has experienced several outbreaks of DENV and it is the main MBV circulating in the human population and posed a major public health threat. It is likely that the first detection of DENV in West Africa was in Ibadan, Nigeria in 1964 from febrile patients [42]. However, the most recent outbreak of DENV was reported in Ouagadougou, Burkina Faso in 2017 [59]. In addition to DENV outbreaks in Burkina Faso, there were reports in other countries, such as Nigeria, Côte d’Ivoire, Senegal, Sierra Leone, Benin, Mali, Cape Verde, Ghana, Mauritania, and Guinea in a decreasing order of incidence. Therefore, Burkina Faso had several DENV reports unlike other West African countries. The main vectors for DENV are *Aedes* mosquitoes (mainly *Aedes aegypti*) through which DENV is transmitted to humans (amplifying host). DENV was mainly detected via serological investigations in humans, although *Ae. aegypti* mosquitoes were also documented as a source of detection.

### 3.2. Togaviridae

Members of this family are enveloped, spherical, icosahedral, and 65–70 nm in diameter. They have a capsid with icosahedral symmetry made of 240 monomers. Additionally, they have a monopartite, linear, ssRNA(+) genome of 9.7–11.8 kb [153]. Alphaviruses are mostly transmitted by mosquito vectors and they have amplifying vertebrate hosts. Acute infections in vertebrates are characterised by a high titer of the virus making it possible for mosquitoes to be infected in the course of blood feeding [154].

Tai Forest alphavirus (TFAV), Eilat virus (EILV), Agua Salud alphavirus (ASALV), and Mwinilunga alphavirus (MWAV) are the main mosquito-specific members of this family [134,155]. However, only TFAV was identified and isolated in West Africa. TFAV was isolated from *Culex decens* mosquitoes collected in 2004 in Côte d’Ivoire [134]. TFAV has a relationship with western equine encephalitis virus complex through phylogenetic analyses [134]. 

Medically important mosquito-borne alphaviruses that are emerging public health threats include Venezuelan (VEEV) and eastern (EEEV) equine encephalitis viruses, and CHIKV. These viruses can cause outbreaks of severe meningoencephalitis with frequent lethal consequences, or polyarthritis with agonizing and chronic joint pain [154]. Likely, the first epidemic of CHIKV in West Africa occurred in 1969 in Ibadan, Nigeria [105]. It is probable that the virus was first isolated from *Aedes* mosquitoes (*Ae. aegypti*, *Ae. grahami*, and *Ae. taylori*) collected in 1963 in Lagos, Nigeria, and serological survey revealed antibodies in the sera of study participants [104]. The recent detection of CHIKV in West Africa occurred between 2016 and 2017 in Sierra Leone and this was via Ion Torrent semiconductor sequencing to profile pathogen spectrum in archived human yellow fever virus-negative sera [110]. A broader spectrum of pathogens was suggested by the distribution of sequencing reads and must be considered in clinical diagnostics and epidemiological surveillance [110]. Until now, a greater number of cases of CHIKV were reported in Sierra Leone, Nigeria, and Senegal, compared to the very few cases in Ghana, Benin, Mali, and Liberia. In general, CHIKV was mainly detected by serology in humans, although there were a few *Aedes* mosquito detections.

### 3.3. Peribunyaviridae

These viruses are enveloped, spherical, or pleomorphic virions 80–120 nm in diameter. They have a segmented, linear ssRNA(-) genome, with L segment between 6.8 and 12 kb, M segment between 3.2 and 4.9, and S segment between 1 and 3 kb [156]. This is a newly established family which includes four genera for globally distributed viruses: Orthobunyavirus, Herbevirus, Pacuvirus, and Shangavirus. Mostly, peribunyaviruses are maintained in a vertebrate–arthropod transmission cycle [157].

The mosquito-specific members of the family include Ferak virus, Jonchet virus, Herbert virus, and Tai virus. Interestingly, all these MSVs were identified in Côte d’Ivoire [135,136,137]. These viruses were identified via a mosquito surveillance study, where mosquitoes were trapped from February to June 2004 in Taï National Park, Côte d’Ivoire [158]. The mosquito samples were further analysed to identify and isolate these viruses [135,136,137]. However, the viruses were mainly isolated from *Culex* mosquitoes.

The mosquito-borne members of the *Peribunyaviridae* include Zinga virus, Bunyamwera virus, Arumowot virus, Bwamba virus, Ngari virus, Nyando virus, Pangola virus, Akabane virus, and M’poko virus [1,3]. Bunyamwera virus is the prominent MBV of this family. Bunyamwera virus was first detected in 1963 in West Africa, Nigeria, from *Mansonia africana* mosquitoes [109]. The virus was also reported to be detected in humans and mosquitoes in Senegal and Guinea respectively [79,83]. 

### 3.4. Other Virus Families

An increasing number of MAVs, especially the mosquito-specific ones, are distributed globally. MAVs were also classified under the following virus families: *Rhabdoviridae, Mesoniviridae, Totiviridae, Reoviridae, Phenuiviridae, Permutotetraviridae, Iflaviridae*, *Xinmoviridae, Nodaviridae*, and Negeviruses (proposed taxon). The latest mosquito-specific mesonivirus is Dianke virus, which was isolated from mosquitoes in Eastern Senegal [143]. Interestingly, Dianke virus was identified in different species of mosquitoes, including *Aedes* sp, *Culex* sp, *Mansonia* sp, and *Uranotaenia* sp. This explains why mesoniviruses are said to have a broad host range and geographical distribution. Another recent and novel mosquito-specific iflavirus isolated from *Aedes vexans* mosquitoes in Senegal is Aedes vexans iflavirus [149]. *Ae. vexans* mosquito is a competent vector of numerous MBVs, such as RVFV and ZIKV. Therefore, it is possible that Aedes vexans iflavirus could interfere with the replication and transmission potentials of RVFV and ZIKV. This review therefore suggests studies involving the interaction of MSVs with MBVs to answer the search for a probable biological vector control tool against arboviruses. 

The medically important mosquito-borne member of the family *Phenuiviridae*, with several reports, is the RVFV. RVFV was reported in Burkina Faso, Sierra Leone, Guinea, Mali, Nigeria, Senegal, and Mauritania. It is likely that the first report of RVFV was in 1959 in Nigeria [126]. However, the latest RVFV case was detected in 2016 in Mali, Niger, and Nigeria [122,125,127], and this is the first occurrence of RVFV in Niger. RVFV was detected in culicine mosquitoes and serological investigations in humans, bats, livestock, horses, and camels.

## 4. Vectors of Mosquito-Associated Viruses

Mosquitoes that play an important role in the transmission of MAVs could be found in the subfamilies of *Anophelinae* and *Culicinae*. Most of world’s mosquitoes belong to the *Culicinae* and species are distributed over all continents, except Antarctica. Females can be distinguished from *Anophelinae* by the short palps and the trilobed scutellum. Legs and scutum often have characteristic scaling and setae patterns, and wings are often broader than those of *Anophelinae* [152]. Highly effective vectors can be found within the *Culicinae* genera *Aedes* and *Culex*. *Aedes* mosquitoes have a highly diverse morphology, but common characteristics are the absence of prespiracular setae and the presence of postspiracular setae.

### 4.1. Vectors of the Genera Aedes

The two most prominent species that transmit viruses are *Ae. aegypti* and *Ae. albopictus. Ae. aegypti* is a medium-sized dark species which can be recognised by its lyre-shaped markings on the scutum. The proboscis is dark-scaled and the clypeus has white scale patches. This species is found in tropical, subtropical, and warm temperate regions of the world. The eggs are resistant to desiccation and are deposited close above the water surface. Larvae occur in a wide variety of small artificial and natural containers with breeding water of a low or moderate content of organic matter. At 27–30 °C, adults emerge 9–10 days after egg laying. In urban environments, females are often found resting indoors and feeding on humans. They are frequent and aggressive daytime biters and do not migrate over long distances. Its evolutionary origin lies in sub-Saharan Africa, from where it was brought to the New World in the 16th century via slave trade ships [153]. The spread to Southeast Asia and the Pacific may have occurred later in the 19th century [153]. The ancestral subspecies named *Aedes aegypti formosus* (Aaf) could still be found in forest ecotones of sub-Saharan Africa and is characterised through its markedly darker appearance. It is a generalist tree hole breeder feeding on animal hosts, and never has pale scales on the first abdominal tergite. The paler domesticated subspecies *Aedes aegypti aegypti* (Aaa) is found in close association with human habitats, breeds in artificial containers (e.g., tyres and tin cans), and primarily bites humans [154]. This form is found outside of Africa and has been responsible for most of the human diseases transmitted by this species [155]. Population structure studies show that, nowadays, the two subtypes have started to interbreed and that, especially in African cities, the situation is not that clear-cut anymore as a result of extensive current or recent historical gene flow. Hybrids of the two forms can be found, as well as domesticated forms of the formerly sylvatic Aaf [156]. The evolutionary adaptations of the domesticated subspecies are particularly important when looking at its role as a vector for arboviruses. With the domestication of vector species, including a change in the host-feeding behaviour (zoophagy to anthropophagy), humans are challenged with new pathogens previously confined to animals. McBride et al. documented striking divergence in preference for human versus animal odour in the two forms, which is linked to an increased expression of an odorant receptor recognising a compound present at high levels in human odour. These host shifts impact the efficiency of mosquitoes as vectors of infectious disease [157]. The most dynamic situation with regard to domestication of *Ae. aegypti* is occurring in West Africa. There are multiple independent incidents of sylvan populations moving into cities. With the expansion of human habitats and cities, an independent domestication is occurring where formerly sylvan mosquitoes are moving into cities [158]. *Ae. aegypti* is known to transmit DENV, YFV, CHIKV, ZIKV, and Mayaro virus [152,159]. It is assumed to be a vector of Venezuelan equine encephalitis virus [160], and vector competence studies have shown *Ae. aegypti* is capable of transmitting WNV [161]. Traditionally, populations identified as Aaf have been considered less competent for DENV and YFV [162]. Recent studies show that it is very difficult to make any general assumptions; vector competence seems to be extremely dependent upon viral and vector strains [163,164,165].

Another mosquito species of great medical importance is *Ae. albopictus*. This species can be recognised by its acrostichal stripe of narrow white scales on the scutum and its bare and entirely dark clypeus. In former times, *Ae. albopictus* was distributed in the Oriental region and Oceania. In Europe, it was first described in Albania in 1979 [166], and in the US in 1985 [167]. *Ae. albopictus* was transported to many countries of the world and has undergone a dramatic global expansion through passive transport of eggs in used tyres or lucky bamboo. Today it can be found in North America, Central America and Caribbean, Asia, some countries in Europe, and South America. In Africa, it was first detected in 1989 in South Africa [168], and eggs were found in 1991 in forested areas of south-central Nigeria [169]. Establishment of populations was reported in 2000 from Cameroon [170]. Afterwards, it has been identified in Equatorial Guinea, Mali, Côte d’Ivoire, Ghana, Democratic Republic of the Congo, Republic of the Congo, Sao Tome, and Principe. Eggs of *Ae. albopictus* are resistant to desiccation and therefore can be transported over long distances. The immature stages occur in a wide variety of artificial and natural containers. In tropical countries, they breed throughout the year, whereas in temperate regions like Europe, populations can be found that show embryonic diapause during winter. Females mostly feed on humans, but other mammals like rabbits, dogs, cows, and squirrels could also be found as hosts. They cause a great nuisance feeding during daytime outside houses, and in dwellings during dusk and night [152]. Under laboratory conditions, *Ae. albopictus* is a competent vector for 29 arboviruses; from field collected specimens, the following arboviruses could be isolated: DENV-1 to 4, WNV, JEV, CHIKV, EEEV, Potosi virus, Cache Valley virus (CVV), Tensaw virus, Keystone virus, La Crosse virus (LCV), Jamestown Canyon virus (JCV), and USUV [171,172]. In Africa DENV-2 and CHIKV have been isolated from field-caught *Ae. albopictus* [173]. In areas where *Ae. albopictus* and *Ae. aegypti* co-occur, both species often share the same larval habitats. In central Africa, *Ae. albopictus* was found breeding in man-made containers, mostly tyres, with vegetation around. *Ae. aegypti* preferred larval habitats located in a neighbourhood with high building density [174]. Specimens of *Ae. aegypti* were more present in the early rainy season, whereas *Ae. albopictus* was most abundant in the late rainy season. This could be due to the greater desiccation tolerance of *Ae. aegypti* eggs [175]. *Ae. albopictus* is a more competent vector for the transmission of DENV, CHIKV, and probably ZIKV in central Africa [176]. Several outbreaks suggest an epidemiological modification of arboviral diseases in the central African region as *Ae. albopictus* was established as the major vector, dominating *Ae. aegypti*. In West Africa, reports of *Ae. albopictus* are patchy, but suitable environments have been predicted using ecological models [2].

Other *Aedes* species play an important role as bridge vectors or in the transmission in sylvatic cycles. Arbovirus species like DENV, YFV, CHIKV, and RVFV were isolated from *Aedes furcifer*, a tree hole breeder that is found in the Afrotropical region [41,43,60,98,112]. This species is known to feed on nonhuman primates and serves as a bridge vector which transmits sylvatic DENV to humans [177]. *Aedes luteocephalus* is another vector that transmits sylvatic DENV and YFV in nonhuman primates. DENV-2 was isolated from arboreal *Aedes luteocephalus* mosquitoes in eastern Senegal, in a relatively remote collection site far from human habitations [39]. Moreover, *Aedes vittatus* and *Aedes taylori* were found to be infected with DENV in Senegal [41,43], and YFV was detected in *Aedes metallicus* collected in a forest gallery in Burkina Faso [98]. During the rainy season in Africa, such sylvatic vectors can reach high densities in gallery forests and moist savannah regions. They can be responsible for rapid virus amplification and spillover into humans. Once the virus is present in humans, interhuman transmission can be sustained and amplified by the domesticated vectors *Ae. aegypti* or *Ae. albopictus* [177].

### 4.2. Vectors of the Genera Culex

In the genus *Culex* several species serve as vectors of one or more important diseases of birds, humans, and other animals. Members of this genus are usually small- to medium-sized species with sparse pleural scaling. The scutellum is distinctly trilobed, prespiracular setae are absent, and the abdomen is blunt-ended with short, oval cerci. Females lay eggs in rafts of as many as 300 on the water surface. Suitable habitats for egg laying vary from small bodies of standing water to large collections of polluted wastewater. *Culex quinquefasciatus* is a member of the *Culex pipiens* complex and is the most abundant species in tropical Africa. *Cx. quinquefasciatus* larvae have been reported in several types of habitat, including clear water, brackish, polluted water with organic matter and human waste, ditches, sewage, latrines, and artificial containers. It is one of the most troublesome mosquitoes biting humans, endophagic during the night and exophagic from sunset to dawn. They frequently feed on birds or other domestic animals, including dogs, cats, and pigs [152]. *Cx. quinquefasciatus* is a known arbovirus vector for CHIKV, Japanese encephalitis virus (JEV), St. Louis encephalitis virus (SLEV), and western and eastern equine encephalitis virus (WEEV, EEEV), WNV, and USUV. In West Africa, only WNV was detected in *Cx. quinquefasciatus*. To our knowledge, only two studies from West Africa (Senegal and Mauritania) describe the finding of arbovirus infections in *Culex* mosquitoes [83,85]. *Culex* species, like *Culex neavei*, *Culex poicilipes*, *Culex tritaeniorhynchus*, and *Culex antennatus*, were found to be infected with Acado virus, Bagaza virus, Barkedji virus, Ndumu virus, Sanar virus, and Yaounde virus, all viruses of unknown pathogen potential. Sindbis virus, USUV, and WNV could also be detected in *Culex* species and are known to mainly cause mild febrile illness with rash. *Cx. neavei* predominantly feeds on birds, and only to a lesser extent on humans, cattle, and horses. Its vector competence for USUV has been proven [178]. *Cx. antennatus* is a zoophagic vector feeding mainly on domestic animals and humans, and could play an important role as an epizootic vector of WNV [80].

### 4.3. Vectors of the Genera Anopheles

*Anopheles* mosquitoes are the vectors of human malaria, but the *Anopheles* virome is poorly studied, and the number and function of viruses are unknown [14]. Adults of this subfamily do not have any scales at least on the first abdominal tergite, and the palps are approximately of the same length as the proboscis. Larvae have no discernible respiratory siphon and rest under the water surface in a horizontal position [152]. Only one arbovirus is known to be transmitted by *Anopheles* mosquitoes, the alphavirus o’nyong-nyong [14]. Other viruses like Batai virus [179], Japanese encephalitis virus [180], and Myxoma virus [181] have been isolated from *Anopheles*. In Africa, Ndumu virus [83,85], Nyando virus [182], WNV [183], and RVFV [184] were present in *Anopheles* females. It is currently unknown if *Anopheles* can contribute to the transmission and maintenance of arboviruses.

## 5. Abiotic Factors Affecting Arboviruses

Factors associated with arbovirus’ distribution include human population growth and migration, and subsequently environment and climate changes due to human activities. In the tropics, extensive urbanisation seems to be the most important factor, creating favourable habitats that are colonised by the anthropohilic and most competent arbovirus vector, *Ae. aegypti* and other *Aedes* mosquito species [27,185]. Globalisation of international exchanges and global warming are changing the map of MAVs and their vectors’ distribution, bringing pathogens and their vectors to regions where they were absent. The colonisation of the six continents by *Ae. albopictus* and the recent ZIKV outbreak in Brazil, with back spreading to Africa, including West Africa, is still recent. West Africa is recognised as the next hotspot for dengue and other arboviruses [24]. Though some studies on the effect of environment and climate changes exist for Africa, they lack resolution for the West African situation in terms of mosquito and mosquito-associated viruses’ evolution. The factors that affect MAVs and their vectors’ distribution can be divided to urbanisation, land use, human population growth, and climate change.

### 5.1. Effect of Urbanisation on Mosquito Species Diversity and Arboviruses

The sub-Saharan Africa is considered the world’s fastest urbanising region, with 472 million people recently moving into cities, but this number is expected to double over the next 25 years [186]. The rapid urbanisation that is taking place in Africa will drive mosquito host switch to humans, causing a shift to human biting [27]. Several review and few research papers claimed unanimously that rapid urbanisation and climate change will alter arboviral disease patterns, but consistent data from West Africa are still missing. While Zahouli et al. [26] demonstrated that urbanisation is the main driving factor of *Aedes* mosquito by creating more diverse and more productive breeding sites compared to semi-urban and rural zone in Côte d’Ivoire, waste management by the municipality in Ouagadougou is found to be associated with increasing dengue risk in the population [67]. The urbanising cities in West Africa lack sanitation infrastructure, which include waste management systems, thus favouring the proliferation of *Cx. quinquefasciatus*, a ubiquitous mosquito adapted to polluted environments.

### 5.2. Human Population Growth Migration and Arboviruses

Braack et al. [1] concluded that human population growth associated with increased international exchanges will likely increase and sustain the threat and geographical spread of current and new arboviral diseases. sub-Saharan Africa’s population is growing at 2.7% a year, which is more than twice as fast as South Asia (1.2%) and Latin America (0.9%), and the population is estimated to double by 2050 [187]. Accelerated global trade and human population movement facilitated by air travel contributed to virus’ transport to nonimmune populations, as well as facilitating the transfer of mosquitoes to new localities [2]. It is admitted that slave trade in the 16th century was the main spreader of *Ae. aegypti* to the Americas, and later to Europe and Asia [188]. In particular, the Sahel region is subject to instability, with population migration to urban areas which may increase the risk of disease in these populations. The growing population also affects land use, including deforestation for agriculture and food safety.

### 5.3. Effect of Land Use on Mosquito Diversity and Arboviruses

Anthropic changes, including land use for agriculture or urbanisation, have a negative effect on mosquito species diversity. A reduction of mosquito species from 75 to 28 was observed in the urbanising cities of Accra and Tema in Ghana [189]. Deforestation for population settlement, agricultural purposes, or others resulted in changes in vectors’ ecology and behaviour that, in turn, may influence the prevalence, incidence, and distribution of vector-borne diseases [190]. In Senegal, environmental factors, such as vegetation index, distance to forest, landscape path size, and rainfall, are correlated with mosquito abundance and even arbovirus transmission. By modelling the ecological niche partitioning by *Aedes* chikungunya vectors, Richman et al. [191] have highlighted that environmental variables are associated with mosquito distribution, with landscape fragmentation having the strongest effect, and the expansion of agricultural and mining activities is likely to increase the risk of sylvatic arbovirus spillover. Land use for agriculture purposes has a specific effect on *Aedes* mosquito abundance and diversity. The rainforest with the highest *Aedes* diversity has the lowest *Aedes* mosquito abundance, while polyculture localities and villages have the highest *Aedes* mosquito densities [25].

### 5.4. Effect of Climate Variability on Mosquito Diversity and Arbovirus Transmission

It is admitted that global warming affects vectors’ distribution and, consequently, disease distribution. Warming temperatures are likely to promote a better environment for dengue and other arboviruses transmitted by *Ae. aegypti*, while it is expected to decrease malaria incidence in sub-Saharan Africa [33]. Weather extremes resulted in excessive rainfall and flooding, as well as severe drought, which caused 10 to 80% variation in major agricultural commodity production (including wheat, corn, cotton, sorghum) and created exceptional conditions for extensive mosquito-borne disease. Between 2010 and 2011, severe drought coupled with higher temperatures increased water storage in Sudan, which, in turn, increased *Ae. aegypti* population density and, subsequently, dengue cases, while in South Africa, at the same period, exceptional rainfall combined with lower temperature created favourable conditions for the most widespread outbreak of RVFV in the region [192]. Though these phenomena are not specific for West Africa, they give a clear image of the effect of climate instability on arbovirus transmission. Rose et al. [27] pointed out the long dry season as the main driving factor of *Ae. aegypti* adaptation to human habitats and specialisation on human hosts. Deforestation is an increasing anthropogenic force in the West African region; this could therefore lead to extensive distribution of *Ae. aegypti* mosquitoes and the diseases they transmit, which must be considered for future outbreak preparedness and control. 

## 6. Discussion

### 6.1. Summary

West Africa is already highly vulnerable to the multiple emergences and re-emergences of arboviruses infecting domestic animals and humans [1,2,3]. In this review 35 MBVs and 38 MSVs are listed in Table 1. DENV, YFV, CHIKV, ZIKV, RVFV, and WNV predominate among arboviral cases reported from humans [2,3]. Incidences of medically important MBVs, such as DENV and ZIKV, were recently reported in West Africa. Recent occurrence of DENV among humans was reported in Benin, Burkina Faso, and Côte d’Ivoire [54,58,78]. Additionally, ZIKV was recently described to be in circulation causing acute infection among humans in Mali and Nigeria [92,94]. Regarding the nonpathogenic MSVs, the most classified ones (e.g., Gouleako and Nounane virus) were identified in Côte d’Ivoire. However, Odorna virus and Aedes vexans iflavirus were recently identified in Ghana and Senegal respectively [136,149].

In West Africa, data on MSVs and their mosquito and vertebrate host ranges are still sparse. Data on MSVs are available only from a few studies and few countries, namely Senegal, Côte d’Ivoire, Ghana, Burkina Faso, and Liberia. However, a larger number of data are available on MBVs detected in wild and domestic animals, and humans with infection cases more concentrated in urban areas and areas subjected to strong climate change effects. The urbanised environments derived from the destruction and the disturbance of natural ecosystems can result in segregation of larval habitats (natural to artificial) and hosts (animal to human), and adaptive domestication of *Ae. aegypti* advantaging more domestic, anthropophilic, and competent subspecies (i.e., *Aaa*) over sylvan and zoophilic subspecies (i.e., *Aaf*) [27,193,194]. *Cx. quinquefasciatus* belong to the *Cx. pipiens* complex that can segregate under anthropogenic changes. As a result, recent arboviral outbreaks have emerged and become more concentrated in urban areas [2,3], where the main vectors *Ae. aegypti* and *Cx. Quinquefasciatus* and the newly-documented *Ae. albopictus* are abundant [25]. Uncontrolled urbanisation provides more breeding sites and higher human population density, offering greater blood feeding opportunities to these highly anthropophagic adult mosquitoes, while climate warning accelerates mosquito larval development. This increases the number and the density of adult vectors, human-biting rates, and thus exacerbates the arbovirus transmission risk [33]. Additionally, the rapid increase in international trade, tourism, travel, and mobility driven by globalisation highly represent a potential risk of exposure and exportation of these disease vectors and their associated pathogens to new regions, and worldwide [27,195]. Human-biting *Ae. aegypti* is an important disease vector which originally evolved as a byproduct of breeding in artificial and discarded containers (e.g., tyres and tin cans) during rainy season [26], and human-stored water in urban areas to survive the long, hot dry season [26,27]. Rose et al. model also predicts that the extensive urbanisation (e.g., world population is projected to reach 9.8 billion in 2050 and 11.2 billion in 2100 [196]) currently taking place in Africa will drive further evolution and spread of MAVs and their mosquito vectors, causing a shift toward human biting in many large cities by 2050 [27]. Climate change may be contributing to human population migration to urban areas and rapid urbanisation that may, in turn, increase the abundance of solid waste and the need for water storage that can potentially provide larval habitats for *Aedes* mosquitoes, and increase the vector densities and arbovirus transmission [27,197]. Improved understanding of these pathways is important for characterising spatial and temporal distributions of arboviral risk and defining arboviral risk calculations, and their interactions with meteorological variables and human behaviours [3]. There is an urgent need for building and strengthening research and interventional capacities for arboviral disease and vector surveillance to reflect change in the epidemiology of these diseases in the region. To fill these key information and knowledge gaps, it should be better to define the spatial and temporal patterns of arboviral disease risk, and formulate effective control plans to respond to this crucial public health challenge [3,27,198].

### 6.2. Major Knowledge Gaps, Challenges, and Future Research Directions

Presently, transmission cycles of MBVs in West Africa are unclear, as the available information about the primary mosquito vectors, vertebrate hosts, and the ecology of these MAVs are very sparse. Moreover, to understand the full range of MAV characteristics, and factors of their emergences and re-emergences in their original foci and their global propagation, it is necessary to know the relationship to the original strains that have been circulating in Africa for a longer period, and their characteristics. It is also important to understand the transmission dynamics, and the impacts of the biological interactions at playon vector and host ecologies, dispersal, and adaptation to new territories. The mechanism of triggering and the emergence of arboviruses in West Africa still remains largely obscure; local arboviruses are apprehended through case reports, often during current epidemics, or exported cases outside Africa, i.e., on other continents (e.g., Asia and Europe). The small quantity of data available on MSVs and their potential impacts on the transmission of MBVs may not be explained by their scarcity in nature, but by the lack or little number of dedicated studies and resulting data due to the critical limitations of specific resources, technical capacities, and budgets directed to this research area [2,3]. Thus, failure to detect arboviral risk early before the occurrence of related outbreaks might result from poor vector and arbovirus surveillance due to very poor investments and limited scientific, technical, and operational capacities [2,3]. Another important gap is a disconnection and absence or insufficiency of concerted actions between researchers, stakeholders at the national arbovirus control programs, and the targeted or local communities, thus posing the problem of sustainability of the implementation of outbreak responses and routine or evidence-based interventions, and their impacts [199,200,201]. To address these crucial challenges, the following urgent priorities are recommended: strengthening the capacity of laboratories, technicians, researchers in West Africa to detect MAVs; training, empowering, and involving local policymakers, practitioners, and communities, with gender-based analysis and action component in vector control activities in a holistic approach to sustain the interventions; addressing current knowledge gaps in our understanding of possible change in the ecology *Aedes* mosquito vectors in the region; developing innovative tools for early detection of MBVs, early warning systems, rapid diagnostics, and prevention strategies derived from the MSV capacities to block arbovirus transmission; and building efficient, cost-effective, and integrated vector control programs and local and regional virus and vector surveillance networks to enhance collaborations between laboratories in West Africa. Additionally, the local researchers and stakeholders and decision makers in West Africa should be encouraged to collaborate with their international homologues, mainly from the arbovirus endemic countries in America, Asia, and Europe. Such international cooperation with skilled and competent experts can help with transferring new knowledge, skills, and expertise to improve or build innovative vector and arbovirus research, surveillance, and control plans in West Africa. As arboviral outbreaks occur mostly in urban areas, the *Aedes* vector control strategy should be integrated and based on multi-sectorial approaches involving policy, community engagement, urban planning, vector control, and environmental sanitation and hygiene sectors for solid, plastic water management, wastewater, and potable water adduction among households’ impacts [199,200,201]. Moreover, it should be recommended to explore, in the light of *Ae. aegypti* behavioural patterns (i.e., biting, blood feeding, and resting behaviours), the possibility of using or modifying the existing *Anopheles* malaria vector tools and methods (e.g., insecticidal long-lasting net and indoor residual spraying) to control this arbovirus vector. Indeed, there are important financial investments and well-established technical competencies in malaria research and control domain in West Africa. The increase in dengue outbreaks and the introduction of *Ae. albopictus* can lead to an interrogation of the effectiveness of the traditional *Stegomyia* indices (i.e., house index, container index, and Breteau index) to measure accurately arboviral risk, as these risk indices have been initially defined to estimate *Ae. aegypti* larval indices and the risk of transmission of YFV in Africa [202,203,204]. In this new context, it should be recommended to formulate an alternative calculation of dengue risk indices in Africa. Moreover, particular attention should be drawn on febrile illnesses in West Africa as arboviral infections are often recorded or misdiagnosed as malaria [205,206,207]. Therefore, it should be important to “deconstruct malaria-industrial complex [24,208]” by performing additional tests to check the presence of an arbovirus among acute fever cases in inpatients and outpatients in the local hospitals [24,208,209,210,211,212,213]. Finally, the authors propose the incorporation of arbovirus testing into the routine laboratory investigations in the West African healthcare setting, especially in the arboviral endemic areas; this should also be encouraged anytime there is a suspicion (early warning signal) of an outbreak.

## 7. Conclusions

West Africa has experienced a profound shift in the epidemiology of MAVs and the acceleration of MBVs that have caused considerable veterinary, public, and global health concerns. During this last decade, the spread of MBVs has resulted in a drastic increase in the emergence and re-emergence of urban arboviral outbreaks (27,000 cases since 2017 [3]) threatening chiefly citizens in the western African region. Indeed, large and multiple outbreaks of DENV, YFV, CHIKV and ZIKV have recently occurred in highly urbanised areas, mainly in major West African cities. The sero-prevalence surveys principally focus on clinical cases in inpatients and outpatients in local hospitals. Moreover, huge numbers of arboviral cases are recorded as nonmalaria febrile illnesses or misdiagnosed as malaria in West Africa. Arboviral infections are sometimes reported as exported arboviral cases outside West Africa (i.e., Asia, America, and Europe). This suggests that the true burden of the diseases is underestimated and far higher than case reports. The data available on MSVs are sparse, but indicate high diversity and abundance of these viruses in the areas where they are assessed. This implies that there are still major gaps in their distribution and impacts on the ecology of mosquito vectors and the transmission of arboviruses. Both urban vectors *Ae. aegypti* and *Cx. quinquefasciatus*, and the recently-introduced *Ae. albopictus* are important vectors. Large human population size due to fast urbanisation and warming temperatures driven by climate change may provide, respectively, great blood-feeding opportunities and alter habitat suitability for arboviral vectors, and exacerbate the risk of transmission of arboviruses and arboviral disease outbreaks in the region. This review identifies critical information gaps related to the definition of the spatial and temporal patterns of *Aedes* mosquito vectors and arboviral outbreak risk, and to the formulation of effective control strategies to respond to arboviral outbreaks with huge veterinary, public, and international health dimensions. To fill the knowledge gaps, there is a pressing need to build scientific, technical, and operational capacities for improving the research and control of arboviruses and vector surveillance to reflect the changing epidemiology of arboviral diseases in West Africa.

## Figures and Tables

**Figure 1 viruses-13-00891-f001:**
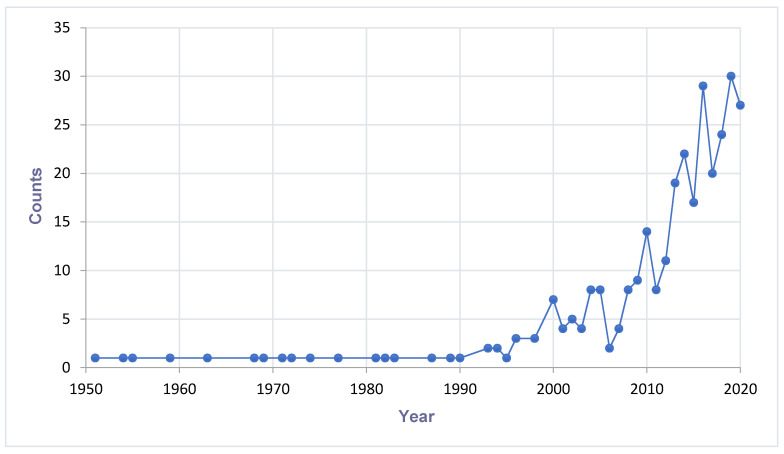
Trend of mosquito-associated viruses reported in literature in West Africa.Increasing trend of mosquito-associated virus discovery and research (Count = combination of virus discoveries and researches on mosquito-associated viruses). A search at the NCBI database (https://www.ncbi.nlm.nih.gov) by entry of the syntax ‘mosquito-borne viruses, West Africa’, and ‘mosquito specific viruses, West Africa’. Last search date: 8 December 2020.

**Figure 2 viruses-13-00891-f002:**
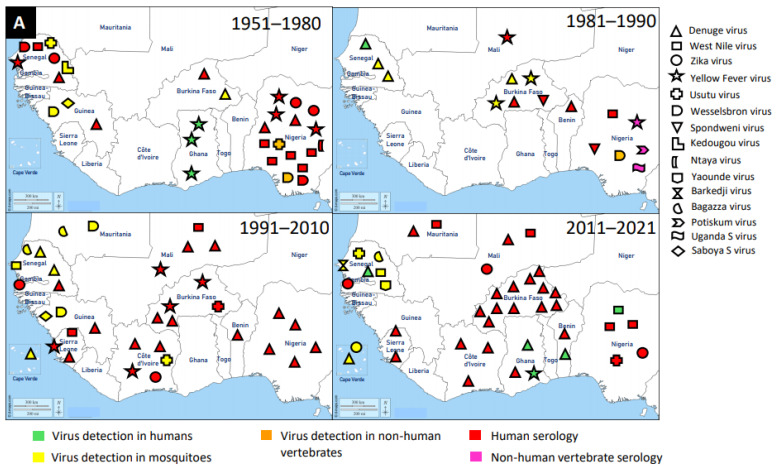
Distribution of MAVs in West Africa. A–C Mosquito-borne viruses (**A**) Flaviviridae; (**B**) Togaviridae; (**C**) Peribunyaviridae, Rhabdoviridae and Reoviridae; (**D**) Mosquito-specific viruses. Location of the signs in the country was chosen randomly and does not refer to the location where the study was performed; * = Proposed taxon; ** = Unclassified viruses.

**Table 1 viruses-13-00891-t001:** Mosquito-associated viruses (MAVs) discovered in West Africa from 1951–2021.

Family	Name of Virus	Year of Virus Detection	Country	Source of Virus Detection	Reference
***Flaviviridae***	**Mosquito-borne viruses**				
	**Dengue**	2009	Cape Verde	*Aedes aegypti*	[34]
		2014/15	Cape Verde	*Ae. aegypti*	[35]
		2016/17	Ghana	Serology/Human	[36]
		2016/17	Ghana	Human	[37]
		2014/2016	Ghana	Serology/Human	[38]
		1974	Senegal	*Ae. luteocephalus*, Serology/Human	[39]
		1983	Senegal	Human	[40]
		1990	Senegal	*Aedes* sp	[41]
		2015/19	Senegal	Human	[42]
		1999	Senegal	*Aedes* sp	[43]
		2009/10	Senegal	*Ae. aegypti,* Serology/Human	[44]
		1964	Nigeria	Serology/Human	[45]
		1975	Nigeria	Serology/Human	[46]
		2001	Nigeria	Serology/Human	[47]
		2010 *	Nigeria	Serology/Human	[48]
		2011	Nigeria	Serology/Human	[49]
		2013 *	Nigeria	Serology/Human	[50]
		2014 *	Nigeria	Serology/Human	[51]
		1987/93	Benin	Serology/Human	[52]
		2010	Benin	Serology/Human	[53]
		2019	Benin	Serology/Human; Human	[54]
		2015	Mauritania	Serology/Human	[55]
		2017	Burkina Faso	Serology/Human	[56]
		2016	Burkina Faso	Serology/Human	[57]
		2019	Burkina Faso	Serology/Human	[58]
		1980	Burkina Faso	*Aedes* sp	[59]
		1983/86	Burkina Faso	*Aedes* sp	[60]
		2013/14	Burkina Faso	Serology/Human	[61]
		2015/17	Burkina Faso	Serology/Human	[62]
		2014	Burkina Faso	Serology/Human	[63]
		2016/17	Burkina Faso	Serology/Human	[64]
		2016	Burkina Faso	Serology/Human	[65]
		1982	Burkina Faso	Serology/Human	[66]
		2004	Burkina Faso	Serology/Human	[67]
		2016	Burkina Faso	Serology/Human	[68]
		2013/14	Burkina Faso	Serology/Human	[69]
		2017	Burkina Faso	Serology/Human	[56]
		2003/4	Burkina Faso	Serology/Human	[70]
		2016	Burkina Faso	Serology/Human	[71]
		2009/13	Mali	Serology/Human	[72]
		2006	Mali	Serology/Human	[73]
		1999	Côte d’Ivoire	Mosquitoes, Serology/Human	[74]
		2010	Côte d’Ivoire	Serology/Human, Human	[75]
		2011/12	Côte d’Ivoire	Serology/Human, Human	[76]
		2017	Côte d’Ivoire	Serology/Human, Human	[77]
		2019	Côte d’Ivoire	Serology/Human	[78]
	**Dengue**	1978/91	Guinea	Mosquitoes	[79]
		2006/8	Sierra Leone	Serology/Human	[80]
		2016 *	Sierra Leone	Serology/Human	[81]
		2012/13	Sierra Leone	Serology/Human	[82]
	**West Nile**	2012/13	Senegal	*Culex* sp, *Aedes* sp	[83]
		1972/75	Senegal	Serology/Human	[84]
		1998/99	Senegal	*Masonia uniformis*	[85]
		1951/55	Nigeria	Serology/Human	[86]
		1968/69	Nigeria	Serology/human	[87]
		1963	Nigeria	Serology/Human	[88]
		1975	Nigeria	Serology/Human	[89]
		1987	Nigeria	Serology/Horses	[90]
		2011/12	Nigeria	Serology/Horses	[91]
		2018	Nigeria	Human	[92]
		2015	Mauritania	Serology/Human	[55]
		2009/13	Mali	Serology/Human	[72]
		2006/8	Sierra Leone	Serology/Human	[80]
	**Zika**	2016	Cape Verde	*Ae. aegytpi*	[93]
		2016	Mali	Serology/Human	[94]
		1972/75	Senegal	Serology/Human	[84]
		1999	Côte d’Ivoire	Mosquitoes, Serology/Human	[74]
		2010/14	Gambia	Serology	[95]
		1975 *	Nigeria	Serology	[46]
		1951/55	Nigeria	Serology/Human	[86]
		2018	Nigeria	Serology	[92]
	**Yellow Fever**	1989	Nigeria	Serology/horses	[90]
		1951/55	Nigeria	Serology/Human	[86]
		1968/69	Nigeria	Serology/Human	[87]
		1975	Nigeria	Serology/Human	[89]
		1972/75	Senegal	Serology/Human	[84]
		1976	Senegal	*Aedes* sp	[96]
		2003/8	Burkina Faso	Serology/Human	[97]
		1983/86	Burkina Faso	*Aedes* sp	[60]
		1983	Burkina Faso	*Aedes* sp	[98]
		1999	Burkina Faso	Serology/Human	[99]
		1987	Mali	Serology/Human	[100]
		2006	Mali	Serology/Human	[73]
		1999	Côte d’Ivoire	Mosquitoes, Serology/Human	[74]
		2006/09	Sierra Leone	Serology/Human	[80]
		1977/80	Ghana	Human	[101]
		1963	Ghana	Human	[101]
		1969/70	Ghana	Human	[101]
		2011	Ghana	Human	[102]
	**Usutu**	2012/13	Senegal	*Culex* sp	[83]
		1972/1977	Senegal	*Aedes* sp	[103]
		1972	Nigeria	*Turdus libonyanus*	[104]
		2018	Nigeria	Serology/Human	[92]
		2004	Burkina Faso	Serology/Human	[104]
		2004	Côte d’Ivoire	*Culex quinquefasciatus*	[104]
	**Wesselsbron**	1972/75	Senegal	Serology/Human	[84]
		1971 *	Nigeria	Serology/Horse	[105]
		1975	Nigeria	Serology/Human	[89]
		1989 *	Nigeria	Serology/Horse	[90]
		1998	Mauritania	*Aedes vexans*	[85]
		1978/91	Guinea	Mosquitoes	[79]
	**Spondweni**	1982 *	Burkina Faso	Serology/Human	[106]
	**Kedougou**	1978 *	Senegal	*Aedes* sp	[107]
	**Ntaya**	1977	Nigeria	Serology/Human	[108]
	**Yaounde**	2012/13	Senegal	*Culex* sp	[83]
	**Bagazza**	2012/13	Senegal	*Culex* sp	[83]
		1998/99	Senegal	*Aedes fowleri*	[85]
		1998/99	Mauritania	*Culex neavei*	[85]
	**Barkedji**	2012/13	Senegal	*Culex* sp, *Aedes* sp	[83]
	**Potiskum**	1989 *	Nigeria	Serology/Horses	[90]
	**Uganda S**	1989 *	Nigeria	Serology/Horses	[90]
	**Saboya**	1978/91	Guinea	Mosquitoes	[79]
***Togaviridae***					
	**Chikungunya**	1963	Nigeria	*Aedes* sp	[109]
		1969	Nigeria	Serology/Human	[110]
		1968/69	Nigeria	Serology/Human	[87]
		1989 *	Nigeria	Serology/Horse	[90]
		1974	Nigeria	Serology/Human	[111]
		2009/10	Senegal	Serology, *Aedes* sp	[112]
		1972/75	Senegal	Serology/Human	[84]
		2016/17	Ghana	Serology/Human	[36]
		2006	Benin	Serology/Human	[113]
		2009/13	Mali	Serology/Human	[72]
***Togaviridae***					
	**Chikungunya**	2012/13	Sierra Leone	Serology/Human	[114]
		2016/17	Sierra Leone	Serology/Human	[115]
		2006/8	Sierra Leone	Serology/Human	[80]
		2012/13	Sierra Leone	Serology/Human	[82]
		1975/77	Sierra Leone	Serology/Human	[108]
		1977	Liberia	Serology/Human	[108]
	**Semliki Forest**	1971	Senegal	Serology/Horses	[116]
		1951/55	Nigeria	Serology/Human	[86]
		2014 *	Nigeria	Serology/Human	[117]
	**Sindbis**	2012/13	Senegal	*Culex* sp	[83]
		1972/75	Senegal	Serology	[84]
	**Ndumu**	2012/13	Senegal	Several mosquitoes	[84]
	**Onyongnyong**	1989 *	Nigeria	Serology/Horses	[90]
		1974	Nigeria	Serology/Human	[108]
		1975	Nigeria	Serology/Human	[108]
		1954	Ghana	Serology/Human	[108]
		1975/77	Sierra Leone	Serology/Human	[108]
***Phenuiviridae***					
	**Rift Valley fever**	1993/96	Burkina Faso	*Aedes* sp	[60]
		1987	Burkina Faso	Serology/Sheep	[118]
		2005/7	Burkina Faso	Serology/livestock	[119]
		1985/87	Burkina Faso	Serology/livestock	[120]
		2006/8	Sierra Leone	Serology/Human	[80]
		1987 *	Guinea	Serology/Bats	[121]
		1978/91	Guinea	Mosquitoes	[79]
		2016	Mali	Serology/Human	[122]
		2015	Mali	Serology/Human	[123]
		2005/14	Mali	Serology/Bovine	[124]
		2016	Niger	Serology/Human/ Livestock	[125]
		1959	Nigeria	Serology/Sheep	[126]
		2016	Nigeria	Serology/Camels	[127]
		1989	Nigeria	Seroloy/Horses	[90]
		1998	Senegal	*Culex* sp, Serology	[128]
		2012/13	Senegal	*Aedes ochraceus*	[83]
		1998/99	Senegal	*Culex poicilipes*	[85]
		1998	Mauritania	*Culex poicilipes*	[85]
		2015	Mauritania	Serology/Human	[55]
***Peribunyaviridae***					
	**Zinga**	1975/1977	Nigeria	Serology/Human	[108]
	**Bunyamwera**	1978 *	Senegal	Serology/Human	[84]
		1951/55	Nigeria	Serology/Human	[86]
		1963	Nigeria	*M. africana*	[109]
		1978/91	Guinea	Mosquitoes	[79]
	**Arumowot**	1968/69	Nigeria	Serology/Human	[87]
	**Bwamba**	1969/72	Nigeria	Serology/Human	[129]
		1951/55	Nigeria	Serology/Human	[86]
	**Ngari**	2010	Mauritania	Serology/Goat	[130]
	**Nyando**	1972 *	Senegal	Serology/Human	[131]
	**Pongola**	1963	Nigeria	*M. africana*	[109]
	**Akabane**	2015	Nigeria	Serology/Livestock	[132]
	**M’Poko**	1978/91	Guinea	Mosquitoes	[79]
***Rhabdoviridae***					
	**Mossuril**	1978/91	Guinea	Mosquitoes	[79]
***Reoviridae***					
	**Acado**	2012/13	Senegal	*Culex sp*	[83]
	**Sanar**	2012/13	Senegal	*Culex neavei, M. uniformis*	[83]
		1998/99	Senegal	*Culex poicilipes*	[85]
	**Kindia**	1978/91	Guinea	Mosquitoes	[79]
	**African horse sickeness**	1971 *	Nigeria	Serology/Horse	[105]
		1993 *	Nigeria	Serology/Camels/Donkey/ Dogs/Horses	[133]
		1993 *	Nigeria	Serology/Horse	[134]
		1995 *	Nigeria	Serology/Horse	[135]
***Flaviviridae***	**Mosquito-specific viruses**				
	Culex flavivirus	2016	Ghana	*Culex* sp	[136]
	Cell fusing agent virus	2016	Ghana	*Aedes aegypti*	[136]
	Anopheles flavivirus-like 2	2012	Senegal	*Anopheles* sp	[13]
	Anopheles flavivirus-like 1	2012	Senegal	*Anopheles* sp	[13]
	Nienokoue	2004	Côte d’Ivoire	*Culex* species mosquitoes	[137]
	Nounane	2004	Côte d’Ivoire	*Uranotaenia mashonaensis*	[138]
***Togaviridae***					
	Taï Forest alphavirus	2004	Côte d’Ivoire	*Culex decens*	[139]
***Peribunyaviridae***					
	Ferak	2004	Côte d’Ivoire	*Culex decens*	[140]
	Jonchet	2004	Côte d’Ivoire	*Culex* sp	[140]
	Herbert	2004	Côte d’Ivoire	*Culex nebulosus*	[141]
	Tai	2004	Côte d’Ivoire	*Culex* sp	[141]
***Rhabdoviridae***					
	Moussa	2004	Côte d’Ivoire	*Culex decens*	[142]
***Mesoniviridae***					
	Odorna	2016	Ghana	*Aedes aegypti*	[136]
	Dianke	2013	Senegal	*Culex poicilipes*	[143]
	Cavally	2016	Ghana	*Aedes aegypti*	[136]
	Cavally	2004	Côte d’Ivoire	*Aedes harrisoni*	[144]
	Nse	2004	Côte d’Ivoire	*Culex nebulosus*	[145]
	Meno	2004	Côte d’Ivoire	*Uranotaenia chorleyi*	[145]
	Hana	2004	Côte d’Ivoire	*Culex* sp	[145]
	Moumo	2004	Côte d’Ivoire	*Culex* sp	[145]
***Totiviridae***					
	Aedes aegypti totivirus	2016	Ghana	*Aedes aegypti*	[136]
***Reoviridae***					
	Aedes pseudoscutellaris reovirus	2015	Ghana	*Aedes aegypti*	[136]
	Cimodo	2008	Côte d’Ivoire	*Culex nebulosus*	[146]
***Phenuiviridae***					
	Phasi Charoen-like phasivirus	2016	Ghana	*Aedes aegypti*	[136]
	Gouleako	2004	Côte d’Ivoire	*Culex* sp	[147]
**Negeviruses ****					
	Dezidougou	1984	Côte d’Ivoire	*Aedes aegypti*	[148]
***Iflaviridae***					
	Aedes vexans iflavirus	2017	Senegal	*Aedes vexans*	[149]
***Permutotetraviridae***					
	Culex permutotetra-like virus	2016	Ghana	*Culex* sp	[136]
***Nodaviridae***					
	Mosinovirus	2004	Côte d’Ivoire	*Culicidae*	[150]
***Xinmoviridae***					
	Bolahun virus variant 2	2012/15	Liberia	*Anopheles gambiae*	[13]
**Unclassified *****				
Unclassified Riboviria	Bolahun virus variant 1	2012/15	Burkina Faso	*Anopheles gambiae*	[13]
Unclassified Riboviria	Aedes aegypti virga-like virus	2016	Ghana	*Aedes aegypti*	[136]
Unclassified Riboviria	West Accra	2015	Ghana	*Aedes aegypti*	[136]
Unclassified Riboviria	Mole Culex	2016	Ghana	*Culex* sp	[136]
Unclassified Riboviria	Goutanap	2004	Côte d’Ivoire	*Culex nebulosus*	[151]
Unclassified Riboviria	Goutanap	2016	Ghana	*Culex* sp	[136]
Unclassified Riboviria	Tesano Aedes	2016	Ghana	*Aedes aegypti*	[136]
Unclassified Riboviria	Korle-bu Aedes	2016	Ghana	*Aedes aegypti*	[136]

* = Year of research publication used; ** = Proposed taxon [148]; *** = Unclassified viruses; Year/Year = Study period.

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
