# Peer review of "Mosquito-Associated Viruses and Their Related Mosquitoes in West Africa"

_viruses, 2021, doi:10.3390/v13050891_

Round 1

Reviewer 1 Report

This is a straightforward and detailed review regarding arboviruses and their vectors in West Africa.  It possesses sufficient depth and focus to be impactful to a specialized audience.  I only have a set of minor suggestions to improve the presentation:

Minor Points:

  1. Abstract, line 19:  remove the comma after As
  2. Abstract line 26: remove the first of the two ands between the mosquito species and replace it with a comma
  3. Methods, line 114: The acronym RDT does not appear to follow from ‘Rapid Immunochromatographic Test”.  Perhaps this could be explained for the general reader.
  4. Line 152: change researches to research
  5. Line 161: To be accurate, I would suggested changing, “surface proteins are arranged in an icosahedral-like symmetry” to “surface proteins of the viral capsid are arranged in an icosahedral-like symmetry”  as the envelope proteins that are on the surface of the virus are not symmetrically arranged.
  6. Line 177: remove the = sign
  7. Line 204: for consistency, do not capitalize the ‘A’ in Tai Forest Alphavirus
  8. Line 358: pics = pigs?
  9. Line 466: change raining forest to rain forest?
  10. There is a recent paper by Dieng et al (doi: 10.3390/v13010057) on DENV in Senegal that the authors may wish to discuss to ensure that the manuscript is as up to date as possible.

Author Response

Dear Reviewer,

thanks for the revision of our manuscript. We have changed the manuscript according to your comments and also corrected a few minor typographical errors. Please see the attachment. All changes are highlighted in yellow and track changes were also applied.

Best regards,

Hanna Jöst

Reviewer 2 Report

In this review, the authors summarized the incidence and prevalence of mosquito-borne viruses and mosquito-specific viruses in West Africa, presenting with abiotic factors such as human population, environment and climate changes. To date many medically important arboviruses such as West Nile virus (Uganda), Zika virus (Uganda), Chikungunya virus (Tanzania) and Rift valley fever virus (Kenya) have been reported primary from East Africa, and comprehensive arbovirus researches including non-pathogenic mosquito-specific viruses are required in West Africa. The manuscript is well established, and it will provide a useful information for the arbovirus researchers, physician and veterinarian.

Minor critique:

  1. Page 2, line 62

Do you mean that Rift valley fever virus causes abortion in pregnant women commonly? If so, the number of report is limited, but the following paper support your statement.

Association of Rift Valley fever virus infection with miscarriage in Sudanese women: a cross-sectional study, Baudin M. et al., Lancet Glob Health. 2016 Nov;4(11):e864-e871. doi: 10.1016/S2214-109X(16)30176-0. Epub 2016 Sep 28

  1. Page 7 and 9, Table 1

To date Rift valley fever virus and Gouleako virus belong to Phenuiviridae but not Peribunyaviridae. Please see the following website of International Committee on Taxonomy of Viruses (ICTV).

https://talk.ictvonline.org/taxonomy/

  1. Page 16, line 384

Misspelling…Pigs?

Author Response

(The authors gave the same response as above.)
